# Menstrual health and factors associated with school absence among secondary school girls in Luang Prabang Province, Lao People's Democratic Republic: A cross-sectional study

**Souphalak Inthaphatha**[1], **Viengsakhone Louangpradith**[2], **Leyla Isin Xiong**[3], **Valee Xiong**[3], **Ly Ly**[3†], **Vue Xaitengcha**[3], **Alongkone Phengsavanh**[4], **Nobuyuki Hamajima**[1], **Eiko Yamamoto**[1] *

1 Department of Healthcare Administration, Nagoya University Graduate School of Medicine, Nagoya, Aichi, Japan, 2 Department of Healthcare and Rehabilitation, Ministry of Health, Vientiane Capital, Lao People's Democratic Republic, 3 Days for Girls International, Luang Prabang City, Luang Prabang Province, Lao People's Democratic Republic, 4 Faculty of Medicine, University of Health Sciences, Vientiane Capital, Lao People's Democratic Republic

† Deceased.
* yamaeiko@med.nagoya-u.ac.jp

**Data Availability Statement:** Data cannot be shared publicly because the informed consent that

## Abstract

In Lao People's Democratic Republic (Lao PDR), information on school sanitation and menstrual health among secondary school girls is limited. This study aimed to explore knowledge and practices surrounding menstrual health and to identify factors associated with school absence due to menstruation among secondary school girls in Lao PDR. The study involved 1,366 girls from grade 9 to grade 12 in six secondary schools in Luang Prabang Province. Data on socio-demographics and menstrual health of the girls and data on school toilets was collected. Logistic regression analysis was performed to identify the factors associated with school absence due to menstruation. The mean age was 15.8 years old. The average age of menarche was 12.9 years old. Of 1,366 girls, 64.6% were shocked or ashamed when they reached menarche and 31.8% had been absent from school due to menstruation in the six months before this study was conducted. Factors associated with school absence due to menstruation were age ≥ 16 years old (AOR = 1.79, 95% CI 1.37–2.34), higher income (AOR = 2.38, 95% CI 1.16–4.87), menstrual anxiety (AOR = 1.55, 95% CI 1.09–2.20), using painkillers (AOR = 4.79, 95% CI 2.96–7.76) and other methods (AOR = 2.82, 95% CI 1.86–4.28) for dysmenorrhea, and disposing used pads in places other than the school's waste bins (AOR = 1.34, 95% CI 1.03–1.75). Living with relatives (AOR = 0.64, 95% CI 0.43–0.95) and schools outside the city (AOR = 0.59, 95% CI 0.38–0.90) were significantly less associated with school absence. Although the association between school toilets and school absence was not examined, the results of this study suggest that school toilets should be gender-separated and equipped with waste bins in the toilet. Furthermore, menstrual education should start at elementary schools and teacher training on menstrual health should be promoted.

was obtained from respondents did not specify that data would be made publicly available and the public availability of data would compromise the privacy of respondents. The data are available upon request to the National Ethics Committee for Health Research (http://www.laohrp.com/index.php/hrp/index) for researchers who meet the criteria for access to confidential data. Researchers who would like to access to the data must contact Administrator of Lao Health Research Portal, Lao Tropical and Public Health Institute, Ministry of Health, Samsenthai Road, Ban Kaognot, Sisattanak District, Vientiane Capital, Lao PDR. Tel: 856-020-55949082, Fax: 856-021-214012 (http://www.laohrp.com/index.php/hrp/about/index/contactlinks).

**Funding:** EY received Research Grant for Asian Region Study by Heiwa Nakajima Foundation in 2021 (http://hnf.jp/). The funders had no role in study design, data collection and analysis, decision to publish, or preparation of the manuscript.

**Competing interests:** The authors have declared that no competing interests exist.

## Introduction

Menstruation is a natural process of the female reproductive system. Girls express their secondary sex characteristics, both physiology and emotions, and change through the adolescent period [1]. Adolescence is a transition period where girls experience cognitive development and social changes take place [2]. It is a crucial part of human life because whether a girl develops into a healthy adult predominantly depends on how she is nurtured during the adolescence period [3].

At menarche, girls often experience negative feelings, including fear, shame, and guilt towards their families [4], because they lack menstrual information and they are not prepared for reaching menarche [5]. Menstrual hygiene practices vary among girls according to different levels of knowledge [6], area of residence [7], disposable (commercial) pad availability [4, 8, 9], and available sanitation facilities [10, 11]. Poor menstrual hygiene management is associated with poor mental health and poor physical health, such as urinary tract infection and bacterial vaginosis [12–15]. Bacterial vaginosis may cause reproductive tract infections and lead to pelvic inflammatory disease [14, 16], and unhygienic practices during menstruation is reported to be one of risk factors of secondary infertility [16, 17]. In addition, poor menstrual health can influence school attendance as well as academic performance [4, 18–20]. In the worst case, girls drop out of school because they cannot manage their menstrual hygiene when they are at school [19, 20].

Female inclusion in education has been prioritized globally because improving girls' education can delay child rearing, increase contraceptive uptake, and improve the nutritional status and vaccination rate of their children when they become mothers [21, 22]. Furthermore, girls who obtain accurate and pragmatic information on sexual and reproductive health are less likely to encounter sexually transmitted diseases, unintended pregnancy, and abuse [23]. However, school environments, including sanitation facilities, are often not supportive due to a lack of adequate water and inadequate toilet facilities [24–26]. Therefore, increasing attention has been paid to improving menstrual hygiene management and integrating menstruation into the Water, Sanitation and Hygiene (WASH) programs to ensure that girls can maintain their school attendance and manage their menstruation with comfort and hygiene while in school [12, 27].

In Lao People's Democratic Republic (Lao PDR), approximately 59% of schools did not have sanitation facilities and 61% of schools did not have improved water source in 2008 [28]. Therefore, many students had to return home to use the toilet or defecate in the forest. WASH programs were first implemented in Lao PDR in 1990. They aim to establish safe water, provide hygiene education, and ensure that Lao people can access adequate sanitation. Through a number of governmental and non-governmental organizations led WASH initiatives, Lao PDR has made significant progress in increasing the use of improved sanitation facilities throughout the country as 73.8% of households had access to improved toilets (either shared or privately owned toilets) in 2017 [29]. The Lao Social Indicator Survey in 2017 reported that 81% of Lao women and girls had private places to wash and change menstrual pads when they were at home [29]. However, this does not mean that Lao girls can properly manage their menstruation and access adequate sanitation facilities at school. In addition, it was reported that maintaining good menstrual hygiene was difficult for girls in rural areas in Savannakhet Province and in the northern part of Lao PDR [30]. Therefore, this study aimed to explore knowledge and practices surrounding menstrual health and to identify factors associated with school absence due to menstruation among secondary school girls in Luang Prabang Province, which is located in the northern part of Lao PDR.

## Materials and methods

### Study design and setting

This was a cross-sectional study conducted at six secondary schools in Luang Prabang Province from November 19 to November 27, 2020. Luang Prabang Province consists of 12 districts with a population of 431,889 from diverse ethnic groups [31]. According to a report by the Ministry of Education and Sports, there are 45 public and two private secondary schools (grades 6–12), and 20,851 students (9,606 girls) of grades 9–12 in the province [32]. Six schools were selected from the 45 public secondary schools. Four schools (Santiphab secondary school, Phanluang secondary school, Pongkham secondary school, and Pasathipatai secondary school) were selected from 24 schools in Luang Prabang City (Luang Prabang District) by simple random sampling using a lottery method. Two schools were selected from 21 schools in 11 districts, outside the city, by convenience sampling considering their accessibility; Chomphet secondary school in Chomphet District, and Sobjaek secondary school in Pakxeng District.

### Study population

The study population were secondary school girls of grades 9–12 at the six schools who had reached menarche. Girls who had not reached menarche and who did not provide written informed consent nor parental consent were excluded from the study. Among the 1,515 girls of grades 9–12 in the six schools, 32 girls who had not reached menarche and 117 girls who did not answer all questions were excluded. A total of 1,366 girls were included in the study with a 91.4% questionnaire completion rate among all menstruating girls.

### Data collection instrument and procedure

Data was collected using a structured and self-administered questionnaire. The questionnaire for girls consisted of four sections: socio-demographic information, reproductive information, menstrual knowledge, and menstrual hygiene practice. The questionnaire was developed by retrieving questions from different literatures and was adapted to the Lao cultural context [4, 8, 9, 33, 34]. A pretest was conducted including 30 young females via an online platform to ensure age-level comprehension and a coherent order of questions.

A checklist to assess school toilets was developed based on the results of menstrual hygiene practices in this study, by referring to a previous study [11] and the United Nations Children's Fund (UNICEF) Guidance on Menstrual Health and Hygiene [35]. The information addressed in the checklist included general information about the schools and characteristics of the school toilets. General information composed of the total number of students and girl students, the total number of toilets for students and for girl students, type of toilets, and the cleanliness level of the toilets. The characteristics of the school toilets comprised of gender-separated, distance between male toilets and female toilets, water availability, toilet paper availability, a door latch on each toilet cubicle, light bulbs, waste bins availability, and basin for handwashing. The toilet data was collected using the checklist by an officer of the Luang Prabang Provincial Department of Education and Sports from March 29 to April 2, 2021. The officer was trained in data collection and had previously participated in data collection from girls from November 19 to November 27, 2020.

### Variables

Socio-demographic factors were categorized as follows: the age of the participants was categorized as < 16 years old and ≥ 16 years old based on the mean age (15.8 years old). According to the National Ethnolinguistic Classification, there are four ethnolinguistic groups in Lao

PDR: Lao-Tai, Hmong-Mien, Mon-Khmer and Chinese-Tibetian [29]. However, three ethno-linguistic groups were presented in this study: Lao-Tai, Hmong-Mien, and Mon-Khmer. Three religions were identified in this study: Buddhist, Animism, and Christian. Allowance per week referred to the amount of money that girls received from parents or obtained in a week and was categorized into five groups: 15,000 Lao Kip (LAK), 20,000–40,000 LAK, 50,000–100,000 LAK, 110,000–200,000 LAK, and more than 200,000 LAK (1 USD = 9,453 LAK, as of June 4, 2021). Parental education was classified into five levels: no education, primary school, lower secondary school, upper secondary school, and diploma level or higher. Residential information was obtained to identify where and with whom girls lived and responses were divided into three groups: living with parents, staying with relatives, and staying at a dorm/rental apartment.

Reproductive information was collected to understand the girls' experiences related to menstruation. Age of menarche was classified into three groups based on the mean age of menarche (12.9 years old): < 13 years old, 13 years old, and > 13 years old. Sources of information regarding menstruation prior to the onset of menarche were divided into: none (no information obtained), mothers, sisters, teachers, friends, and others. The shock and feeling of shame that girls had at the onset of menarche referred to their mental preparation for menarche, and was categorized into two groups (yes and no). Menstrual absorbents were divided into four choices: disposable pads, reusable pads, cloth from old towel/skirts, and combined usage of disposable and reusable pads. Disposable pads in this study referred to any commercial and single use pads that are disposed after one use. Dysmenorrheal management was classified into five groups based on the answers by girls who had dysmenorrhea: lay on stomach, take pain-killers, drink hot water, others, and nothing (because it is not so painful). Information on school absence was obtained by asking if a student had been absent from the classroom/school due to menstrual anxiety, menstrual leak, and/or dysmenorrhea in the last six months. The answers were categorized into yes, no, and no response. Girls who did not answer about school absence due to menstruation were not included.

Questions to identify the level of menstrual knowledge and menstrual hygiene practices were retrieved from literature reviews [4, 8, 9, 33, 34]. Five questions concerning menstruation were used to evaluate girl's knowledge [33]: 'What is menstruation?', 'What causes menstruation?', 'Which organ is the origin of menstruation?', 'What is the normal menstruation cycle?', and 'In which phase of the menstrual cycle are women most likely to get pregnant if they have sexual intercourse?' Regarding the latter question, answer choices included: menstrual phase (day 1–7 of menstrual cycle), follicular/proliferation phase (day 8–12 of menstrual cycle), ovulation phase (day 14 of menstrual cycle), and luteal phase (day 15–28 of menstrual cycle). Girls who answered 3–5 questions correctly and 1–2 questions were categorized into 'good knowledge' and 'some knowledge', respectively. Girls who did not get any correct answers and those who answered 'don't know' were categorized into 'poor knowledge'.

## Data analysis

The study used Statistical Package for Social Sciences, version 26 (IBM SPSS Inc, Armonk, NY, USA) for data analysis. Descriptive statistics were used to describe the characteristics of participants and their situation. Logistic regression analysis was performed to obtain the odds ratio (OR) and 95% confidence interval (CI). P value < 0.05 was considered as statistically significant.

## Ethical considerations

The study was approved by the National Ethics Committee for Health Research in Lao PDR (No. 066/NECHR). Written informed consent was obtained from all girl who enrolled in the

study. For all participants aged under 18 years old, written informed consent was also obtained from their guardians including parents, relatives, and teachers before the data collection. The data collection team visited the schools and distributed the forms for guardians three days before the data collection.

## Results

### Socio-demographics of the girls

Among the 1,366 participants from the six schools in Luang Prabang Province, the age of girls ranged from 13 to 19 years old and the mean age was 15.8 years old. Most girls were Lao-Tai (66.6%) and Buddhist (68.5%) (Table 1). The majority lived with their parents (77.9%) and the average allowance per week was 20,000–40,000 LAK (39.7%). Regarding the parental education level, most fathers were educated at university (34.0%). On the other hand, more than half of the mothers were educated at lower secondary school or lower (Table 1).

### Reproductive characteristics of the girls

The average age of menarche among the girls was 12.9 years old. Of 1,366 girls, 64.6% were shocked or ashamed when they reached menarche (Table 2). Most girls responded that they had dysmenorrhea (88.9%). Regarding dysmenorrhea management, the majority of girls (41.9%) reported that they managed dysmenorrhea by drinking hot water. Some of them managed it by laying on their stomach (18.6%) and taking painkillers (16.2%). The major source of menstrual information was mothers (46.9%) and 28.2% of the girls did not have any information about menstruation before menarche.

### Menstrual knowledge of the girls

Misconceptions about menstruation existed among the girls in Luang Prabang Province. Most girls (68.4%) believed that menstruation was bad blood that a woman's body shed monthly. While 546 girls (40.0%) answered that it was because a woman's body wanted to shed bad blood out of her system, 11 girls (0.8%) believed that it was caused by disease (Table 3). Of the 1,366 participants, 208 girls (15.2%) knew that women could get pregnant when they had sexual intercourse during the ovulation phase of the menstrual cycle. The majority of girls had 'some knowledge' about menstruation (61.6%) and only 13.9% of girls had 'good knowledge'.

### Menstrual hygiene practices amongst the girls

Most girls reported using disposable pads (96.6%), changing their pads 2–3 times in one menstrual day (67.1%), and having easy access and being able to afford disposable pads (80.9%) (Table 4). Among the girls, 60.1% were satisfied with the frequency they changed their pads. Regarding using the school toilets, 64.6% of girls reported that they tried to avoid using the school toilets and 77.7% reported that they tried to avoid changing their pads at school. When they did change their pads at school, the majority of girls (73.2%) disposed of their used pad in the school waste bins (either inside the school's toilets or waste bins near the school), however, 23.8% of girls kept the used pads in a plastic bag and took them home for disposal (Table 4). There were 434 girls (31.8%) who reported that they had been absent from school due to menstruation during the past six months (Table 4).

### Characteristics of school toilets

In addition to the girls' behavior and use of school toilets, a school toilet assessment was administered at the six schools in order to comprehensively and holistically understand the

**Table 1. Socio-demographic characteristics of secondary school girls in Luang Prabang Province (N = 1,366).**

| Variables | Categories | n (%) |
|---|---|---|
| Age | | |
| | < 16 years old | 567 (41.5) |
| | ≥ 16 years old | 799 (58.5) |
| Ethno-linguistics | | |
| | Lao-Tai | 910 (66.6) |
| | Hmong-Mien | 247 (18.1) |
| | Mon-Khmer | 209 (15.3) |
| Religion | | |
| | Buddhist | 936 (68.5) |
| | Animism | 389 (28.5) |
| | Christian | 41 (3.0) |
| Residence | | |
| | Live with parents | 1064 (77.9) |
| | Live with relatives | 179 (13.1) |
| | Stay at dormitory/rental apartment | 123 (9.0) |
| Area of school | | |
| | Inside Luang Prabang City | 1137 (83.2) |
| | Outside Luang Prabang City | 229 (16.8) |
| Allowance per week | | |
| | < 15,000 LAK | 198 (14.5) |
| | 20,000–40,000 LAK | 427 (31.3) |
| | 50,000–100,000 LAK | 542 (39.7) |
| | 110,000–200,000 LAK | 147 (10.8) |
| | > 200,000 LAK | 52 (3.8) |
| Father's education | | |
| | No education | 61 (4.5) |
| | Primary school | 297 (21.7) |
| | Lower secondary school | 233 (17.1) |
| | Upper secondary school | 311 (22.8) |
| | Diploma or higher | 464 (34.0) |
| Mother's education | | |
| | No education | 166 (12.2) |
| | Primary school | 365 (26.7) |
| | Lower secondary school | 269 (19.7) |
| | Upper secondary school | 312 (22.8) |
| | Diploma or higher | 254 (18.6) |

menstrual experiences and hygiene practices of the girls at school. Four of the six schools did not have gender-separated toilets and none of the schools provided toilet paper in the school toilets (Table 5). Moreover, four schools did not have waste bins available in any toilet blocks or toilet cubicles. Despite this, the cleanliness of the school toilets was acceptable on the assessment days.

## Factors associated with school absence among the girls

A logistic regression analysis was performed to identify factors associated with school absence due to menstruation (Table 6). In this study, we found that higher age (age ≥ 16 years old)

**Table 2. Reproductive characteristics of secondary school girls in Luang Prabang Province (N = 1,366).**

| Variables | Categories | n (%) |
|---|---|---|
| Age of menarche | | |
| | < 13 years old | 467 (34.2) |
| | 13 years old | 477 (34.9) |
| | > 13 years old | 422 (30.9) |
| Feeling shock and/or ashamed at the menarche | | |
| | Yes | 883 (64.6) |
| | No | 483 (35.4) |
| Duration of menstruation | | |
| | 1–2 days | 8 (0.6) |
| | 3–7 days | 1330 (97.4) |
| | ≥ 8 days | 28 (2.0) |
| Regularity of menstruation | | |
| | Yes | 984 (72.0) |
| | No | 382 (28.0) |
| Dysmenorrhea | | |
| | Yes | 1215 (88.9) |
| | No | 151 (11.1) |
| Dysmenorrhea management | | |
| | Lay on stomach | 254 (18.6) |
| | Painkillers | 221 (16.2) |
| | Drink hot water | 573 (41.9) |
| | Others[a] | 76 (5.6) |
| | Nothing, because it is not so painful | 91 (6.7) |
| Source of menstrual information before menarche attainment | | |
| | None | 385 (28.2) |
| | Mother | 640 (46.9) |
| | Sister | 188 (13.8) |
| | Teacher | 67 (4.9) |
| | Friend | 62 (4.5) |
| | Others[b] | 24 (1.8) |

[a]Others include traditional medicine, hot pack on stomach, and/or one or more methods combined.
[b]Others include reading books and searching the Internet.

(Adjusted OR (AOR) = 1.79, 95% CI 1.37–2.34, P < 0.001) and higher allowance (≥ 200,000 LAK per week) (AOR = 2.38, 95% CI 1.16–4.87, P < 0.05) were significantly associated with school absence. Living with relatives (AOR = 0.64, 95% CI 0.43–0.95, P < 0.05) and studying at schools outside the city (AOR = 0.59, 95% CI 0.38–0.90, P < 0.05) were significantly less associated with school absence due to menstruation than living with their parents and schools in the city, respectively. In addition, girls who responded that they were stressed, irritated, and anxious during menstruation were 1.5 times more likely to be absent from school due to menstruation (AOR = 1.55, 95% CI 1.09–2.20, P < 0.05). Girls who experienced dysmenorrhea to the extent that they needed to treat the pain, including painkillers (AOR = 4.79, 95% CI 2.96–7.76, P < 0.001) and other methods (AOR = 2.82, 95% CI 1.86–4.28, P < 0.001), had significantly higher school absence compared to girls who did not use any dysmenorrhea management methods. Furthermore, girls who disposed of their used pads in places other than

**Table 3. Knowledge of menstruation among secondary school girls, Luang Prabang Province (N = 1,366).**

| Variables | Categories | n (%) |
|---|---|---|
| What is menstruation? | | |
| | Bad blood that women shed monthly | 934 (68.4) |
| | Normal bleeding from a woman's body | 297 (21.7) |
| | Disease | 27 (2.0) |
| | Other answers | 8 (0.6) |
| | Don't know | 100 (7.3) |
| Cause of menstruation | | |
| | Hormones | 628 (46.0) |
| | Body wants to shed bad blood | 546 (40.0) |
| | Disease | 11 (0.8) |
| | Other answers | 11 (0.8) |
| | Don't know | 170 (12.4) |
| Origin of menstrual blood | | |
| | Blood vessel | 14 (1.0) |
| | Uterus | 418 (30.6) |
| | Abdomen | 1 (0.1) |
| | Bladder | 8 (0.6) |
| | Birth canal | 623 (45.6) |
| | Don't know | 302 (22.1) |
| Normal menstrual interval | | |
| | 20–30 days | 442 (32.4) |
| | 28–30 days | 291 (21.3) |
| | 30–40 days | 57 (4.2) |
| | Don't know | 576 (42.2) |
| In which phase of the menstrual cycle are women most likely to get pregnant if they have sexual intercourse? | | |
| | Menstrual phase | 131 (9.6) |
| | Follicular/proliferation phase | 116 (8.5) |
| | Ovulation phase | 208 (15.2) |
| | Luteal phase | 28 (2.0) |
| | Don't know | 883 (64.6) |
| Menstrual knowledge | | |
| | Good knowledge* | 190 (13.9) |
| | Some knowledge** | 841 (61.6) |
| | Poor knowledge*** | 335 (24.5) |

*Girls correctly answered 3–5 questions of 5 questions.

**Girls correctly answered 1–2 questions of 5 questions.

***Girls answered all 5 questions incorrectly.

school's waste bins were significantly more likely to be absent from school due to menstruation than the others (AOR = 1.34, 95% CI 1.03–1.75, $P < 0.05$).

## Discussion

The results of this study showed that 31.8% of secondary school girls missed school due to menstruation within the six months before this study and that there were menstrual knowledge gaps among girls in Luang Prabang Province. In addition, girls who were 16 years old or older, who had a higher allowance > 200,000 LAK per week, and who studied at schools in

**Table 4. Menstrual hygiene practices among girls in Luang Prabang Province (N = 1,366).**

| Variables | Categories | n (%) |
|---|---|---|
| Absorbent materials | | |
| | Disposable pad | 1320 (96.6) |
| | Reusable pad | 35 (2.6) |
| | Cloth from old towel/skirt | 7 (0.5) |
| | Disposable and reusable pad combined | 4 (0.3) |
| Frequency of changing pads per day | | |
| | Once a day | 113 (8.3) |
| | 2–3 times a day | 917 (67.1) |
| | 4–5 times a day | 301 (22.0) |
| | More than 5 times a day | 35 (2.6) |
| Disposable pad accessibility | | |
| | Easy to find and affordable | 1105 (80.9) |
| | Difficult due to financial considerations | 168 (12.3) |
| | Difficult due to distance to shops | 8 (0.6) |
| | Difficult due to both financial considerations and distance to shops | 44 (3.2) |
| | Not sure | 41 (3.0) |
| Are you satisfied with your frequency of changing pads? | | |
| | Unsatisfied, it is not enough | 271 (19.8) |
| | Neutral, not satisfied and not dissatisfied | 275 (20.1) |
| | Satisfied | 820 (60.1) |
| Do you avoid using the school toilets? | | |
| | Yes | 883 (64.6) |
| | No | 483 (35.4) |
| Do you avoid changing pads at school? | | |
| | Yes | 1061 (77.7) |
| | No | 305 (22.3) |
| When you change pads at school, where do you throw away the used pads? (multiple answers allowed) | | |
| | Toilet bowl | 75 (5.5) |
| | Waste bins at school (including waste bins in the school's toilets and waste bin in other areas at school) | 1000 (73.2) |
| | Bury (under dirt) | 56 (4.1) |
| | Keep in a plastic bag and bring home | 325 (23.8) |
| | Others[a] | 32 (2.3) |
| | I have never changed pads in school | 64 (4.7) |
| What do you do with your underwear? | | |
| | I wash and use new underwear daily | 1326 (97.1) |
| | Reuse without washing sometimes | 40 (2.9) |
| How do you clean your reusable pad or cloth? | | |
| | Clean water with detergent always | 32 (69.6) |
| | Clean water with detergent sometimes | 3 (6.5) |
| | No response | 11 (23.9) |
| How do you dry your underwear and your reusable pad? | | |
| | Unhidden, expose to sunlight | 443 (32.4) |
| | Hidden, unexposed to sunlight | 646 (47.3) |

(*Continued*)

**Table 4.** (Continued）

| Variables | Categories | n (%) |
|---|---|---|
| | Not sure | 157 (11.5) |
| | No response | 120 (8.8) |
| Have you been absent from school due to menstruation within the last six months? | | |
| | Yes | 434 (31.8) |
| | No | 932 (68.2) |

[a]Others include throwing away pads in open areas, bushes, and others.

**Table 5.** Characteristics of school toilets at six schools in Luang Prabang Province.

| Variables | School A | School B | School C | School D | School E | School F |
|---|---|---|---|---|---|---|
| Number of students | 2960 | 552 | 985 | 675 | 956 | 358 |
| Number of girl students | 1359 | 263 | 542 | 383 | 496 | 152 |
| Number of toilets | 87 | 27 | 20 | 6 | 4 | 14 |
| Number of functionable toilets | 56 | 27 | 20 | 6 | 4 | 14 |
| Number of toilets for girl students | 0 | 10 | 0 | 0 | 0 | 4 |
| Number of students per toilet | 53 | 20 | 49 | 113 | 239 | 26 |
| Gender-separated toilet | No | Yes | No | No | No | Yes |
| Water for cleaning/flushing the toilet bowl | Yes | Yes | Yes | Yes | Yes | Yes |
| Water for cleaning body after using the toilet | Yes | Yes | No | No | No | Yes |
| Toilet paper in the toilet | No | No | No | No | No | No |
| Door latch in each squat toilet | Yes | Yes | Yes | Yes | Yes | Yes |
| Light bulb in the toilet | Yes | Yes | Yes | Yes | No | No |
| Waste bin in each toilet block | No | Yes | Yes | No | No | No |
| Waste bin in the common area of the toilet room | No | Yes | Yes | No | No | No |
| Waste bin in front of (outside) the toilet | Yes | Yes | Yes | No | No | No |
| Waste bin around school area | Yes | Yes | Yes | Yes | Yes | Yes |
| Waste bin cover | No | No | Yes | No | No | No |
| Hook for hanging belongings | No | No | No | No | No | No |
| Soap for washing hands | Yes | Yes | No | Yes | No | Yes |
| Basin for hand washing in or near the toilet | Yes | Yes | Yes | Yes | No | Yes |
| A clear sign instructing girls to dispose of pads in the waste bin | No | Yes | No | No | No | Yes |
| Level of cleanliness of the school toilet | Considered clean | Acceptable | Acceptable | Acceptable | Considered clean | Considered clean |

Luang Prabang City had significantly higher school absence due to menstruation. Girls who lived with relatives had significantly less school absence than girls who lived with their parents. Moreover, girls who chose to dispose of their used pads in places other than school waste bins, girls who used painkillers and other methods to manage their dysmenorrhea, as well as girls who were stressed, irritated, and anxious during menstruation were associated with school absence.

Most secondary school girls (64.5%) reported that they were shocked and/or ashamed when they had their first menstruation. This result may suggest that the girls did not have adequate information to be prepared for their menarche. The Ministry of Education and Sports of Lao PDR includes concise but limited menstrual education in the reproductive health section under the Natural Sciences textbook for secondary school grade 9 [36], where a majority of girls are aged 13–14 years old. In this study, 69.1% of girls reached menarche before grade 9,

**Table 6. Binary and multivariate analysis on school absence due to menstruation.**

| Variables | School absence | | OR (95% CI) | AOR (95% CI) |
|---|---|---|---|---|
| | No (N = 932) | Yes (N = 434) | | |
| | n (%) | n (%) | | |
| Age | | | | |
| < 16 years old | 434 (76.5) | 133 (23.5) | 1 (reference) | 1 (reference) |
| ≥ 16 years old | 498 (62.3) | 301 (37.7) | 1.97 (1.55–2.51)*** | 1.79 (1.37–2.34)*** |
| Ethno-linguistics | | | | |
| Lao-Tai | 619 (68.0) | 291 (32.0) | 1 (reference) | 1 (reference) |
| Hmong-Mien | 162 (65.6) | 85 (34.4) | 1.12 (0.83–1.50) | 1.24 (0.81–1.89) |
| Mon-Khmer | 151 (72.2) | 58 (27.8) | 0.82 (0.59–1.14) | 1.12 (0.72–1.74) |
| Allowance per week | | | | |
| < 15,000 LAK | 144 (72.7) | 54 (27.3) | 1 (reference) | 1 (reference) |
| 20,000–40,000 LAK | 305 (71.4) | 122 (28.6) | 1.07 (0.73–1.56) | 1.07 (0.69–1.66) |
| 50,000–100,000 LAK | 366 (67.5) | 176 (32.5) | 1.28 (0.89–1.84) | 1.27 (0.81–2.00) |
| 110,000–200,000 LAK | 89 (60.5) | 58 (39.5) | 1.74 (1.10–2.74)* | 1.57 (0.89–2.77) |
| > 200,000 LAK | 28 (53.8) | 24 (46.2) | 2.29 (1.22–4.29)* | 2.38 (1.16–4.87)* |
| Father's education | | | | |
| Unable to read/write | 34 (55.7) | 27 (44.3) | 1 (reference) | 1 (reference) |
| Primary school | 207 (69.7) | 90 (30.3) | 0.55 (0.31–0.96)* | 0.64 (0.33–1.26) |
| Lower secondary school | 156 (67.0) | 77 (33.0) | 0.62 (0.35–1.10) | 0.76 (0.37–1.56) |
| Upper secondary school | 212 (68.2) | 99 (31.8) | 0.59 (0.34–1.03) | 0.64 (0.31–1.32) |
| Diploma or higher | 323 (69.6) | 141 (30.4) | 0.55 (0.32–0.95)* | 0.51 (0.25–1.07) |
| Mother's education | | | | |
| Unable to read/write | 102 (61.4) | 64 (38.6) | 1 (reference) | 1 (reference) |
| Primary school | 256 (70.1) | 109 (29.9) | 0.68 (0.46–1.00)* | 0.71 (0.44–1.15) |
| Lower secondary school | 188 (69.9) | 81 (30.1) | 0.69 (0.46–1.03) | 0.76 (0.44–1.31) |
| Upper secondary school | 215 (68.9) | 97 (31.1) | 0.72 (0.49–1.07) | 0.86 (0.48–1.54) |
| Diploma or higher | 171 (67.3) | 83 (32.7) | 0.77 (0.51–1.16) | 0.94 (0.50–1.77) |
| Residence | | | | |
| Live with parents | 719 (67.6) | 345 (32.4) | 1 (reference) | 1 (reference) |
| Live with relatives | 130 (72.6) | 49 (27.4) | 0.79 (0.55–1.12) | 0.64 (0.43–0.95)* |
| Stay at dormitory/rental apartment | 83 (67.5) | 40 (32.5) | 1.00 (0.67–1.50) | 0.96 (0.59–1.57) |
| School area | | | | |
| In Luang Prabang City | 758 (66.7) | 379 (33.3) | 1 (reference) | 1 (reference) |
| Outside Luang Prabang City | 174 (76.0) | 55 (24.0) | 0.63 (0.46–0.88)** | 0.59 (0.38–0.91)* |
| Menstruation knowledge level | | | | |
| Good knowledge | 133 (70.0) | 57 (30.0) | 1 (reference) | 1 (reference) |
| Some knowledge | 567 (67.4) | 274 (32.6) | 1.13 (0.80–1.59) | 1.33 (0.91–1.94) |
| Poor knowledge | 232 (69.3) | 103 (30.7) | 1.04 (0.70–1.53) | 1.45 (0.94–2.23) |
| Mood and emotion during menstruation | | | | |
| None, I feel no change | 216 (79.1) | 57 (20.9) | 1 (reference) | 1 (reference) |
| Stress, irritated, and anxious | 716 (65.5) | 377 (34.5) | 2.00 (1.45–2.74)*** | 1.55 (1.09–2.20)* |
| Dysmenorrheal management | | | | |
| No pain or mild pain (use nothing) | 207 (85.5) | 35 (14.5) | 1 (reference) | 1 (reference) |
| Painful, took painkillers | 121 (54.8) | 100 (45.2) | 4.89 (3.13–7.63)*** | 4.79 (2.96–7.76)*** |
| Painful, used other methods[a] | 604 (66.9) | 299 (33.1) | 2.93 (1.99–4.30)*** | 2.82 (1.86–4.28)*** |
| Do you try to avoid using toilets at school? | | | | |
| No | 345 (71.4) | 138 (28.6) | 1 (reference) | 1 (reference) |

*(Continued)*

**Table 6.** (Continued)

| Variables | School absence | | OR (95% CI) | AOR (95% CI) |
|---|---|---|---|---|
| | No (N = 932) | Yes (N = 434) | | |
| | n (%) | n (%) | | |
| Yes | 587 (66.5) | 296 (33.5) | 1.26 (0.99–1.61) | 1.19 (0.88–1.60) |
| Do you try to avoid changing pads at school? | | | | |
| No | 220 (72.1) | 85 (27.9) | 1 (reference) | 1 (reference) |
| Yes | 712 (67.1) | 349 (32.9) | 1.27 (0.96–1.68) | 1.16 (0.82–1.64) |
| Type of absorbent materials | | | | |
| Disposable pads only | 901 (68.3) | 419 (31.7) | 1 (reference) | 1 (reference) |
| Others[b] | 31 (67.4) | 15 (32.6) | 1.04 (0.56–1.95) | 0.98 (0.45–2.15) |
| Frequency of changing pads | | | | |
| Two times or more | 852 (68.0) | 401 (32.0) | 1 (reference) | 1 (reference) |
| One time only | 80 (70.8) | 33 (29.2) | 0.88 (0.57–1.34) | 0.82 (0.50–1.32) |
| Mode of pad disposal (for disposable pad users) | | | | |
| Only throw in waste bins at the school (both inside and outside school toilet) | 539 (70.1) | 230 (29.9) | 1 (reference) | 1 (reference) |
| Others[c] | 340 (66.8) | 169 (33.2) | 1.17 (0.92–1.48) | 1.34 (1.03–1.75)* |
| Commercial pad accessibility | | | | |
| Easily accessible | 750 (67.9) | 355 (32.1) | 1 (reference) | 1 (reference) |
| Some difficulties | 182 (69.7) | 79 (30.3) | 0.92 (0.68–1.23) | 0.91 (0.65–1.28) |

OR, odds ratio; AOR, adjusted odds ratio; CI, confidence interval.

[a]Other methods refers to traditional medicines, hot pack on the stomach, and/or two or more methods combined.

[b]Others refers to reusable pads, cloths, or disposable pads and reusable pads combined.

[c]Others refers to throwing in the toilet bowl, bringing home, burning, and throwing away pads in open areas, bushes, and others.

*P < 0.05,

**P < 0.01,

***P < 0.001.

9,453 LAK = 1 USD as of June 4, 2021.

and only 4.9% obtained precise menstrual information from teachers at school before their menarche. The gap between the menstrual education provision and girl's age at menarche might result in fear when they reached menarche. In addition, the percentage of girls who had 'good knowledge' about menstruation was not higher in girls of grades 10–12 (13.2%) compared to that of girls of grade 9 (18.4%). This demonstrated that girls did not fully understand about menstruation even after they were taught at school. Accurate, sufficient, and timely information is necessary for girls to reduce the fear, shame, and stigma related to menstruation, as well as to encourage their good hygiene practice. Menstrual health should be included in the curriculum of grade 5 (10–11 years old) at elementary school, because only elementary education (grades 1–5) is compulsory for all Lao children [37]. Furthermore, teacher training on reproductive health, especially menstrual health, should be promoted and strengthened to ensure that teaching techniques and contents are well integrated and delivered.

School toilets might be a factor affecting school absence due to menstruation for girls in Luang Prabang Province. In this study, girls from schools in Luang Prabang City were associated with school absence. Girls in the city might feel uncomfortable and lack confidence in practicing good hygiene where school toilets did not meet their needs. Among 64 respondents in this study who admitted that they had never changed pads at school toilets at all, 55 girls (85.9%) went to schools in the city, and 9 girls (14.1%) went to schools outside the city. This was consistent with a systematic review which reported that inadequate sanitation facilities

posed challenges for girls to manage their menses with dignity in academic environments [38]. In Lao PDR, the percentage of schools that had basic water supply and sanitation facilities was reported to have improved from 28% in 2000 to 74% in 2017 [39], but there has been no information regarding facilities for menstrual hygiene management and gender-segregated toilets at schools [40]. In this study, four of six schools did not have gender-separated toilets, and did not have waste bins in each toilet block nor in the toilet cubicle. Girl students might feel discomfort and boy students might find out and tease them in the toilet for menstruating [4, 11, 18]. Improved school toilets refer to not only functional toilets but also facilities that are gender-friendly and have necessary items for girls to practice good hygiene [35].

Apart from school toilets, it is reported that dysmenorrhea and mental wellbeing during menstruation were significantly associated with school absence [4, 41]. In this study, girls who took painkillers were 4.79 times more likely to be absent from school, while those using other methods to manage dysmenorrhea were 2.82 times more likely to be absent from schools compared to those who had mild symptoms or did not have dysmenorrhea at all. Painkillers may indicate the severity of the pain where girls cannot conduct their routine activities, including attending school [41, 42]. Qualitative studies in Africa reported that girls were anxious about menstrual leaks onto their clothes and that they had to always be careful and think about it, causing them stress and resulting in them paying less attention to their studies which in turn affected their academic performance [4, 18]. Menstrual hygiene at school should be promoted and menstrual pads and painkillers should be freely available when needed in order to facilitate girls' participation and sustain their educational performance.

In this study, girls with a higher allowance were more likely to be absent from school due to menstruation than those who had a lower allowance. This is inconsistent with many studies in Africa, which revealed that a lower income household was associated with school absence [4, 9, 18]. Low economic status is also reported as a major risk factor of poor menstrual hygiene management and associated with less use of absorbents and disposable pads in other studies [12, 18, 34, 43]. The inconsistency might be due to menstrual context differences, the condition of school toilets, and the proportion of sanitary pad availability and accessibility in different study settings. Previous studies showed that approximately 35% of girls in Western Ethiopia used disposable pads [4], while only 9% of girls in rural Uganda used disposable absorbents [18]. Reusable pad users and cloth users in Africa might struggle to manage menstruation at schools. In this study, on the other hand, 96.6% of girls used disposable pads and 80.9% of girls could easily access and afford disposable pads. The difference of disposable pad usage between girls in Luang Prabang City (98.3%) and outside the city (88.2%) was small, although Goli et al. reported a geographical disparity in accessing sanitation napkins in India [43]. The mass disposable pad usage in this study might contribute to the relationship between girls' allowance and school absence. This suggests that school absence due to menstruation might not be directly linked to socio-economic status in Lao PDR.

This study has some limitations. First, the cross-sectional data cannot identify any causal effect relationship among the factors. Second, the questionnaire required self-reported absenteeism, and some girls might be reluctant to answer this question honestly or might not remember clearly. As 117 girls who did not respond to the question about school absence were excluded from the analysis, the prevalence of school absenteeism in this study might be underestimated. Third, due to the nature of a multiple-choice questionnaire, the true nature of the experiences and practices among girls might not be shown, such as their reasons behind school toilet avoidance. Lastly, we included only girls who could attend secondary school and most of them had easy access to and used disposable pads. Further studies that include girls out of school and use qualitative research methods are necessary to broaden the understanding regarding the challenges in managing menstrual hygiene among Lao girls.

## Conclusion

In this study, the proportion of school absence due to menstruation among secondary school girls in Luang Prabang Province was 31.8%. Factors associated with school absence due to menstruation (age $\geq$ 16 years old, higher income, schools in the city, menstrual anxiety, dysmenorrhea, and disposing used pads in places other than school's waste bins) suggest that school absence due to menstruation might not be directly linked to socio-economic status rather than feeling uncomfortable and a lack confidence in practicing good hygiene at school toilets. Although the association between school toilets and school absence was not examined, the results of this study suggest that school toilets should be gender-separated and equipped with essential waste bins in the toilet. Furthermore, menstrual education should start at elementary school and teacher training on reproductive health, especially menstrual health, should be promoted.

## Acknowledgments

The authors would like to thank to the Ministry of Education and Sports of Lao PDR and the officers from Luang Prabang Provincial Education and Sports Department in facilitating the data collection for this research, especially Ms. Soutsaiychai Douangsavanh for her advice. We would like to thank Ms. Kristen Shalosky and the staff of the Lao Rugby Federation for their collaboration in helping develop the questionnaires during the pretest. Our deep gratitude also goes to all study participants who volunteered and provided valuable information. Finally, we would like to express our condolences to the family of our co-author Ms. Ly Ly. Her contribution and dedication to this research is deeply appreciated.

## Author Contributions

**Conceptualization:** Souphalak Inthaphatha, Leyla Isin Xiong, Eiko Yamamoto.

**Data curation:** Souphalak Inthaphatha, Viengsakhone Louangpradith, Valee Xiong, Ly Ly, Vue Xaitengcha.

**Formal analysis:** Souphalak Inthaphatha, Eiko Yamamoto.

**Funding acquisition:** Eiko Yamamoto.

**Methodology:** Souphalak Inthaphatha, Leyla Isin Xiong, Eiko Yamamoto.

**Project administration:** Souphalak Inthaphatha.

**Supervision:** Nobuyuki Hamajima, Eiko Yamamoto.

**Writing – original draft:** Souphalak Inthaphatha, Eiko Yamamoto.

**Writing – review & editing:** Souphalak Inthaphatha, Viengsakhone Louangpradith, Leyla Isin Xiong, Valee Xiong, Ly Ly, Vue Xaitengcha, Alongkone Phengsavanh, Nobuyuki Hamajima, Eiko Yamamoto.

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
