## [Decision Letter · Decision Letter 0]

31 Aug 2021

PONE-D-21-23049

Menstrual health and factors associated with school absence among secondary school girls in Luang Prabang Province, Lao People’s Democratic Republic: A cross-sectional study

PLOS ONE

Dear Dr. Yamamoto,

Thank you for submitting your manuscript to PLOS ONE. After careful consideration, we feel that it has merit but does not fully meet PLOS ONE’s publication criteria as it currently stands. Therefore, we invite you to submit a revised version of the manuscript that addresses the points raised during the review process.

Considering my own reading and reviewers opinion, I am recommending a minor revision for this paper.  Along with reviewers suggestions, I recommend authors to interpret their findings in comparative perspective with global literature to reach the global readers. I have provided a few references to consult in this regard. 

Malhotra, A., Goli, S., Coates, S., & Mosquera-Vasquez, M. (2016). Factors associated with knowledge, attitudes, and hygiene practices during menstruation among adolescent girls in Uttar Pradesh. Waterlines, 277-305.

Goli, S., Sharif, N., Paul, S., & Salve, P. S. (2020). Geographical disparity and socio-demographic correlates of menstrual absorbent use in India: A cross-sectional study of girls aged 15–24 years. Children and Youth Services Review, 117, 105283.

We look forward to receiving your revised manuscript.

Kind regards,

Srinivas Goli, Ph.D.

Academic Editor

PLOS ONE

Journal Requirements:

Additional Editor Comments:

Considering my own reading and reviewers opinion, I am recommending a minor revision for this paper. Along with reviewers suggestions, I recommend authors to interpret their findings in comparative perspective with global literature to reach the global readers. I have provided a few references to consult in this regard.

Malhotra, A., Goli, S., Coates, S., & Mosquera-Vasquez, M. (2016). Factors associated with knowledge, attitudes, and hygiene practices during menstruation among adolescent girls in Uttar Pradesh. Waterlines, 277-305.

Goli, S., Sharif, N., Paul, S., & Salve, P. S. (2020). Geographical disparity and socio-demographic correlates of menstrual absorbent use in India: A cross-sectional study of girls aged 15–24 years. Children and Youth Services Review, 117, 105283.

Reviewers' comments:

Reviewer's Responses to Questions

**Comments to the Author**

1. Is the manuscript technically sound, and do the data support the conclusions?

Reviewer #1: Yes

Reviewer #2: Partly

Reviewer #3: Yes

2. Has the statistical analysis been performed appropriately and rigorously? 

Reviewer #1: Yes

Reviewer #2: Yes

Reviewer #3: Yes

3. Have the authors made all data underlying the findings in their manuscript fully available?

Reviewer #1: No

Reviewer #2: Yes

Reviewer #3: Yes

4. Is the manuscript presented in an intelligible fashion and written in standard English?

Reviewer #1: Yes

Reviewer #2: Yes

Reviewer #3: Yes

5. Review Comments to the Author

Reviewer #1: I appreciate the opportunity to review this manuscript on menstrual health and factors associated with school absence among secondary school girls in Luang Prabang Province, Lao People’s Democratic Republic. This is a very interesting and well written paper, providing much needed evidence on effect of menstruation on school going girls. I recommend the paper for publication. However, I have few observations which I think the authors can incorporate in the final stage.

• The authors may highlight the important factors associated with school absence due to menstruation with statistical significant values (Odds) in the abstract.

• Use of different types of menstrual absorbent materials could have been also related to school absence. Therefore, it can be included in the regression model.

• The authors should include the limitation of self-reported absenteeism at the end of Discussion section.

Reviewer #2: 1. A brief introduction about the needs and availability/unavailability of the facilities in schools would depict the introduction in a better way.

2. The methodology is not clear. The information on how the schools were selected is missing.

3. How the consent was sought is not clear. How the consent was sought from parents is not clear.

4. “Each parent’s education was classified into five levels”- There is no need to write ‘each parent’s education’ drop the word ‘each.’ (See line 125)

5. Line 146-147 states – “Questions to identify the level of menstrual knowledge and menstrual hygiene practices 147 were retrieved from literature reviews [4, 8, 9, 30].” None of the four references are from the local context. It would be better if authors could include a few local study for citation, if available.

6. Line 147-150. Authors describe about the five questions related to menstruation. This is just a query- whether these questions are self-derived or adopted from somewhere else. Please clarify.

7. It is suggested to use the same terminology everywhere in the manuscript. Authors have used girls/student interchangeably.

8. Conclusion section seems more of the result section. This needs more insights.

Reviewer #3: Overall comment: The study explores the menstrual knowledge and menstrual hygienepractices among schools girls. Most important aspects surrounding this area of research has been covered. Improvement on certain points will make the article publishable.

Introduction: Literature can be added on fertility association with menstrual health, and also menstrual hygiene related diseases.

Materials and Methods: The sampling procedure needs to be added, how the authors have selected the six schools from 45 schools. The dependent and independent variable can be segregated in the variable description part.

Line 174: The authors have considered statistical significance at 5% as written in the data analysis section, but further in results part they have also considered significance at 1% and 0.1% as well. So that can be mentioned too in the data analysis section.

6. PLOS authors have the option to publish the peer review history of their article (what does this mean?). If published, this will include your full peer review and any attached files.

Reviewer #1: No

Reviewer #2: No

Reviewer #3: **Yes: **Sampurna Kundu

---

## [Author Response · Author response to Decision Letter 0]

12 Oct 2021

We would like to thank the editor and the reviewers for reviewing our manuscript. We have revised the manuscript according to your comments, which were very helpful. The revised manuscript has been proofread by a native English speaker. The revisions have been completed and the responses are as follows.

Editor

1. Considering my own reading and reviewer’s opinion, I am recommending a minor revision for this paper. Along with reviewers’ suggestion, I recommend authors to interpret their findings in comparative with global literature to reach the global readers. I have provided a few references to consult in this regard.

- Malhotra, A., Goli, S., Coates, S., & Mosquera-Vasquez, M. (2016). Factors associated with knowledge, attitudes, and hygiene practices during menstruation among adolescent girls in Uttar Pradesh. Waterlines, 277-305.

- Goli, S., Sharif, N., Paul, S., & Salve, P. S. (2020). Geographical disparity and socio-demographic correlates of menstrual absorbent use in India: A cross-sectional study of girls aged 15–24 years. Children and Youth Services Review, 117, 105283.

Thank you very much for your recommendation. After reviewing both papers of the studies conducted in India, we have used the results of these papers to compare with the results of our study as follows. 

Lines 351-353: “Low economic status is also reported as a major risk factor of poor menstrual hygiene management and associated with less use of absorbents and disposable pads in other studies [12, 18, 34, 43].”

Line 359-362: “The difference of disposable pad usage between girls in Luang Prabang City (98.3%) and outside the city (88.2%) was small, although Goli et al reported the geographical disparity in accessing sanitation napkins in India [43].”

Reviewer #1:

1. The authors may highlight the important factors associated with school absence due to menstruation with statistical significant values (Odds) in the abstract.

Lines 38-45: We have added AOR and 95% CI for factors associated with school absence in the abstract as follows, “Factors associated with school absence due to menstruation were age ≥ 16 years old (AOR = 1.79, 95% CI 1.37-2.34), higher income (AOR = 2.38, 95% CI 1.16-4.87), menstrual anxiety (AOR = 1.55, 95% CI 1.09-2.20), using painkillers (AOR = 4.79, 95% CI 2.96-7.76) and other methods (AOR = 2.82, 95% CI 1.86-4.28) for dysmenorrhea, and disposing used pads in places other than the school’s waste bins (AOR = 1.34, 95% CI 1.03-1.75). Living with relatives (AOR = 0.64, 95% CI 0.43-0.95) and schools outside the city (AOR = 0.59, 95% CI 0.38-0.90) were significantly less associated with school absence.”

2. Use of different type of absorbent materials could have been also related to school absence. Therefore, it can be included in the regression model.

Table 6: We appreciate this comment and we understood that absorbent materials should be included in the regression models. Factors associated with school absence due to menstruation were not changed by adding the variable “type of absorbent materials” in binary and multivariate logistic regression analyses. We have revised accordingly Table 6.

3. The author should include the limitation of self-reported absenteeism at the end of Discussion section.

Lines 366-370: A limitation of self-reported absenteeism has been added. “Second, the questionnaire required self-reported absenteeism, and some girls might be reluctant to answer this question honestly or might not remember clearly. As 117 girls who did not respond to the question about school absence were excluded from the analysis, the prevalence of school absenteeism in this study might be underestimated.”

Reviewer #2:

1. A brief introduction about the needs and availability/unavailability of the facilities in school would depict the introduction in a better way.

Lines 79-81: Information on the available facilities in schools in Laos was added as follows, “In Lao People’s Democratic Republic (Lao PDR), approximately 59% of schools did not have sanitation facilities and 61% of schools did not have improved water source in 2008 [28]. Therefore, many students had to return home to use the toilet or defecate in the forest.”

2. The methodology is not clear. The information on how the schools were selected is missing.

Lines 103-109: We are sorry for not clearly explaining how the six schools were selected in the method section. We decided to select four schools in the city and two schools out of the city (in districts). A lottery method was used for random sampling of three schools in the city. However, two schools out of the city were chosen according to easy accessibility including road conditions and the distance from the center of the city to the schools. We have revised the manuscript as follows, “Six schools were selected from the 45 public secondary schools. Four schools (Santiphab secondary school, Phanluang secondary school, Pongkham secondary school, and Pasathipatai secondary school) were selected from 24 schools in Luang Prabang City (Luang Prabang District) by simple random sampling using a lottery method. Two schools were selected from 21 schools in 11 districts, outside the city, by convenience sampling considering their accessibility; Chomphet secondary school in Chomphet District, and Sobjaek secondary school in Pakxeng District.”

3. How the consent was sought is not clear. How the consent was sought from the parents is not clear.

Lines 189-192: To the best of our knowledge, in Lao PDR, people who are 18 or older are legally adults. As shown in Table 1, 22.1% of the students did not live with their parents; 13.1% lived with relatives and 9.0% lived at dormitory/rental apartment. Therefore, in this study, we obtained informed consent from guardians including parents, relatives, and teachers, when girl students were younger than 18 years old. We provided the consent forms for guardians three days before the data collection. We have added as follows, “For all participants aged under 18 years old, written informed consent was also obtained from their guardians including parents, relatives, and teachers before the data collection. The data collection team visited the schools and distributed the forms for guardians three days before the data collection.”

4. “Each parent’s education was classified into five levels” – there’s no need to write ‘each parent’s education’, drop the word ‘each’.

Lines 147-148: We have revised the sentence and dropped the word ‘Each’, as following: “Parental education was classified into five levels: no education, primary school, lower secondary school, upper secondary school, and diploma level or higher.”

5. Line 146-147 states – “Questions to identify the level of menstrual knowledge and menstrual hygiene practices 147 were retrieved from literature reviews [4, 8, 9, 30].” None of the four references are from the local context. It would be better if authors could include a few local studies for citation, if available. 

Lines 168-169: There are not many papers concerning quantitative studies on menstrual knowledge and menstrual hygiene practice in Southeast Asia. When we were plannning this research from late 2019 to early 2020, most of the literature searched by the PubMed library were from Africa and South Asia. Cultures, beliefs, and practices in Africa and South Asia are very different from those in Lao PDR. We cited a paper of a study conducted in Bhutan (previously listed as reference 30) but we have also added a paper from India in the references (reference number 34) as follows, “Questions to identify the level of menstrual knowledge and menstrual hygiene practices were retrieved from literature reviews [4, 8, 9, 33, 34].”

6. Line 147-150. Authors describe about the five questions related to menstruation. This is just a query – whether these question s are self-derived or adopted from somewhere else. Please clarify. 

Lines 169-170: We retrieved the five questions about menstruation knowledge from the publication “Knowledge, attitude and practices of menstrual hygiene management of adolescent school girls and nuns in Bhutan” by the Ministry of Education of Bhutan and UNICEF. This is a report of the survey conducted in Bhutan and explained clearly about the used tool for the survey and the result of the menstrual knowledge using these five questions.We have added the reference number as follows, “Five questions concerning menstruation were used to evaluate girl’s knowledge [33]”

7. It is suggested to use the same terminology everywhere in the manuscript. Authors have used girls/student interchangeably.

We have decided to use “girls” and revised the manuscript. 

8. Conclusion section seems more of the result section. This needs more insights. 

Lines 379-383: We have reduced the summary of the results and added an interpretation of the results in the conclusion section as follows, “Factors associated with school absence due to menstruation (age ≥ 16 years old, higher income, schools in the city, menstrual anxiety, dysmenorrhea, and disposing used pads in places other than school’s waste bins) suggest that school absence due to menstruation might not be directly linked to socio-economic status rather than feeling uncomfortable and a lack confidence in practicing good hygiene at school toilets.”

Reviewer #3:

1. Introduction: Literature can be added on fertility association with menstrual health, and also menstrual hygiene related diseases. 

Lines 60-64: We have added sentences about the association between poor menstrual hygiene management and infertility in the introduction section, citing two more papers as follows, “Poor menstrual hygiene management is associated with poor mental health and poor physical health, such as urinary tract infection and bacterial vaginosis [12-15]. Bacterial vaginosis may cause reproductive tract infections and lead to pelvic inflammatory disease [14, 16], and unhygienic practices during menstruation is reported to be one of risk factors of secondary infertility [16, 17].”

2. Materials and Methods: The sampling procedure needs to be added, how the authors have selected the six schools from 45 schools. The dependent and independent variable can be segregated in the variable description part. 

Lines 103-109: We are sorry for not clearly explaining how the six schools were selected in the method section. We decided to select four schools in the city and two schools out of the city (in districts). A lottery method was used for random sampling of three schools in the city. However, two schools out of the city were chosen according to easy accessibility including road conditions and the distance from the center of the city to the schools. We have revised the manuscript as follows, “Six schools were selected from the 45 public secondary schools. Six schools were selected from the 45 public secondary schools. Four schools (Santiphab secondary school, Phanluang secondary school, Pongkham secondary school, and Pasathipatai secondary school) were selected from 24 schools in Luang Prabang City (Luang Prabang District) by simple random sampling using a lottery method. Two schools were selected from 21 schools in 11 districts, outside the city, by convenience sampling considering their accessibility; Chomphet secondary school in Chomphet District, and Sobjaek secondary school in Pakxeng District.”

Lines 137-179: We have made a new paragraph of “Variables” and moved our explanation about dependent and independent variables from the paragraph of “Data collection instrument and procedure” to the paragraph of “Variables.”

3. Line 174: The authors have considered statistical significance at 5% as written in the data analysis section, but further in the results part they have also considered significance at 1% and 0.1% as well. So that can be mentioned too in the data analysis section. 

Lines 184-185: P value < 0.05 was considered as statistically significant in all analyses of this study. In some tables, we used symbols (** and ***) to understand when P value was < 0.01 and < 0.001 as the same as other papers.

---

## [Decision Letter · Decision Letter 1]

26 Nov 2021

Menstrual health and factors associated with school absence among secondary school girls in Luang Prabang Province, Lao People’s Democratic Republic: A cross-sectional study

PONE-D-21-23049R1

Dear Dr. Yamamoto,

We’re pleased to inform you that your manuscript has been judged scientifically suitable for publication and will be formally accepted for publication once it meets all outstanding technical requirements.

Kind regards,

Srinivas Goli, Ph.D.

Academic Editor

PLOS ONE

Additional Editor Comments (optional):

Considering my own reading and reviewers suggestion, I am going with a decision of accept.

Reviewers' comments:

Reviewer's Responses to Questions

**Comments to the Author**

1. If the authors have adequately addressed your comments raised in a previous round of review and you feel that this manuscript is now acceptable for publication, you may indicate that here to bypass the “Comments to the Author” section, enter your conflict of interest statement in the “Confidential to Editor” section, and submit your "Accept" recommendation.

Reviewer #1: All comments have been addressed

2. Is the manuscript technically sound, and do the data support the conclusions?

Reviewer #1: Yes

3. Has the statistical analysis been performed appropriately and rigorously? 

Reviewer #1: Yes

4. Have the authors made all data underlying the findings in their manuscript fully available?

Reviewer #1: Yes

5. Is the manuscript presented in an intelligible fashion and written in standard English?

Reviewer #1: Yes

6. Review Comments to the Author

Reviewer #1: (No Response)

7. PLOS authors have the option to publish the peer review history of their article (what does this mean?). If published, this will include your full peer review and any attached files.

Reviewer #1: No

---

## [Editor Report · Acceptance letter]

3 Dec 2021

PONE-D-21-23049R1 

Menstrual health and factors associated with school absence among secondary school girls in Luang Prabang Province, Lao People’s Democratic Republic: A cross-sectional study 

Dear Dr. Yamamoto:

I'm pleased to inform you that your manuscript has been deemed suitable for publication in PLOS ONE. Congratulations! Your manuscript is now with our production department. 

Kind regards, 

on behalf of

Dr. Srinivas Goli 

Academic Editor

PLOS ONE